# The Key Role of Mitochondria in Somatic Stem Cell Differentiation: From Mitochondrial Asymmetric Apportioning to Cell Fate

**DOI:** 10.3390/ijms241512181

**Published:** 2023-07-29

**Authors:** Ilario Amato, Sébastien Meurant, Patricia Renard

**Affiliations:** 1Ressearch Unit in Cell Biology (URBC), Namur Research Institute for Life Sciences (Narilis), University of Namur (UNamur), 5000 Namur, Belgium; ilario.amato@student.unamur.be (I.A.); sebastien.meurant@unamur.be (S.M.); 2Mass Spectrometry Platform (MaSUN), Namur Research Institute for Life Sciences (Narilis), University of Namur (UNamur), 5000 Namur, Belgium

**Keywords:** stem cell, differentiation, asymmetric, mitochondria, metabolism, epigenetic, RISP

## Abstract

The study of the mechanisms underlying stem cell differentiation is under intensive research and includes the contribution of a metabolic switch from glycolytic to oxidative metabolism. While mitochondrial biogenesis has been previously demonstrated in number of differentiation models, it is only recently that the role of mitochondrial dynamics has started to be explored. The discovery of asymmetric distribution of mitochondria in stem cell progeny has strengthened the interest in the field. This review attempts to summarize the regulation of mitochondrial asymmetric apportioning by the mitochondrial fusion, fission, and mitophagy processes as well as emphasize how asymmetric mitochondrial apportioning in stem cells affects their metabolism, and thus epigenetics, and determines cell fate.

## 1. Introduction

Stem cells are classified into three categories, according to their differentiation potential. First, embryonic stem cells (ESCs), called pluripotent stem cells (PSCs), can differentiate into all cell types derived from the three embryonic layers. Second, induced pluripotent stem cells (iPSCs), are differentiated cells which have been reprogrammed into PSCs in vitro. Third, somatic stem cells, known as multipotent, can differentiate into the cell types of the tissue from which they originate and are the focus of this review. Stem cell mitosis can be conducted through two different strategies: symmetric and asymmetric cell division. The first one, the symmetric strategy, gives rise to two differentiated daughter cells or two daughter cells retaining the characteristics of stem cells [1] (stem cell renewal and expansion). The asymmetric strategy results in a differentiating cell, plus a cell retaining the stem properties [2]. Original studies, focusing on the role of the stem cell environment in the differentiation process and in the stem cell self-renewal ability, gave rise to the stem cell niche concept, originally proposed by Schofield in 1978 [3]. Since then, numerous articles have shown that multipotent stem cells live in a microenvironment called a “niche” that controls their proliferation and differentiation depending on extracellular cues and thus ensures tissue homeostasis [3,4,5]. The overview of somatic stem cells described in the literature, such as hematopoietic (HSCs), mammary, intestinal, epithelial, or muscle stem cells, show molecular and micro-anatomical dissimilarity in niche controls, depending on the cell type considered [6,7,8,9]. A common feature is that stem cells retain their commitment and behavioral control by maintaining close contact with their niches. On the contrary, to avoid the niche control and to commit to differentiation, the offspring must separate from the key elements composing the niche. This necessary proximity between stem cells and their niche can be explained by the need for molecular exchanges with the environment such as contact with the extracellular matrix or cell–cell interactions [10].

Nevertheless, while the concept of niche helps to explain how the stem cell enters into or avoids mitosis, the niche does not by itself fully explain the cell fate of the progeny and thus all the metabolic and epigenetic rearrangements that occur in the committed cell. The first widespread metabolic reprogramming would be the metabolic shift observed during several differentiation programs (reviewed in [11,12]). Indeed, a preferential glycolytic activity is observed in the stem cell, shifting toward an oxidative phosphorylation activity in the committed/differentiated cell. This shift is supported by a mitochondrial biogenesis occurring during cell differentiation, as observed in multiple differentiation models [13,14,15]. This raises the question of the mitochondrial involvement in stem cell regulation, which was already described to play a role at the metabolic level. Thus, during the last 20 years, scientists investigated the fundamental question of whether the mitochondria-dependent metabolic shift was a cause or consequence of cell differentiation. Two recent studies from the group of Katajisto in 2015 and 2022 suggest that the answer to this cell fate decision could reside in the metabolic impact of the mitochondrial dynamics and their asymmetric apportioning in stem cell [16,17]. This organelle, with pleiotropic functions, such as ATP production or apoptosis regulation and where anabolic and anapleurotic reactions occur, also plays an important part in the metabolism and epigenetic remodeling of stem cells [18].

Over the past 15 years, accumulating epigenetic studies aimed to understand how the same genome, with different epigenetic markers, such as histone acetylation and methylation as well as DNA methylation, can dictate a cell type or its phenotype. The idea is that epigenetic markers determine which gene to transcribe, thus finally defining the transcriptome and the resulting metabolism of the cell [19,20,21,22,23]. However, a new concept has emerged recently, not on how epigenetics shapes the cell but rather on how cellular metabolism changes the epigenetics of the cell and thus influences its phenotype. This inverted control is called cell metabolic reprogramming, and its mechanism in stem cells has just started to be uncovered, with a special emphasis on the role of mitochondria in this process [24,25,26,27]. As a result, the abundance of metabolites affecting epigenetics is modulated by the energy metabolism [28], the cell stage [29], or the type of mitochondria inherited [17] as detailed below. Therefore, histones and DNA modifications operate as a relay between metabolism and cell fate, with modifications of the transcriptome regulated by the abundance of the metabolic intermediates.

The purpose of this review is thus to provide a better understanding of the role played by the mitochondria in stem cell differentiation, with an emphasis on the importance of their asymmetric distribution, the underlying metabolic rewiring, and the resulting epigenetic changes.

## 2. Stemness and Cell Differentiation Are Connected to Mitochondrial Dynamics and Maintenance

Three main types of evidence suggest that stem cell maintenance and division, together with the cell fate of stem cell progeny is closely connected to mitochondrial dynamics and maintenance.

First, the mitochondrial network in stem cells is rather fragmented, while mitochondria in differentiated progeny cell display a dense tubular aspect. This difference suggests that mitochondrial fusion/fission, or more broadly mitochondrial dynamism, is needed and is essential for stemness [30].

Second, several studies show that modulation of mitochondrial dynamics and maintenance in these cells results in major consequences (reviewed in [31]). For instance, the inhibition and the knockdown of the dynamin-related protein 1 (Drp1), the main actor of the mitochondrial fission process, further differentiate iPSCs in cardiomyocytes exhibiting an oxidative metabolic shift and thus implying loss of stemness [32]. On the contrary, forced Drp1 expression promotes iPSC stemness properties [33]. Many other publications report changes in stem cell destiny and stemness properties when the actors of the mitochondrial fusion and fission mechanism (e.g., Drp1, Fis1 or OPA1) are over- or under-expressed [34,35,36,37,38,39].

Third, a variety of data support the importance of autophagy during the differentiation process. Indeed, actively oxidizing mitochondria in stem cells are removed by the mitophagy pathway, probably to preserve stemness [40]. A role for mitophagy in stemness conservation is further supported by the observation that mitophagy deficiencies promote differentiation by leaving mitochondria using primarily oxidative respiration [41]. Differentiation into the myeloid progenitor of HSCs is observed when autophagy is inhibited in these cells. Indeed, the increase in metabolically active mitochondria leads to the metabolic shift characteristic of differentiation [41,42]. Moreover, treatment of bone marrow mesenchymal stem cells (BM-MSCs) with chloroquine and 3-methyladenine, two autophagy inhibitors, results in the inhibition of BM-MSC differentiation [43]. In addition, the impairment of the PINK1–Parkin mitophagy through PINK1 deletion reduces the reprogramming efficiency of somatic cells into iPSCs, further supporting a role of the mitophagy in the stemness maintenance [34].

While it is largely accepted that mitochondria, and more particularly mitochondrial dynamics, play a prominent role in stem cell behavior, it is only recently that a potential contribution of asymmetric apportioning of mitochondria emerged as a new player in stem cell fate determination.

## 3. Asymmetric Mitochondrial Distribution and Stem Cell Fate

Asymmetric distribution of mitochondria in daughter cells following cell division is a phenomenon that has been described in a variety of models, ranging from yeast (reviewed in [44]) to mammalian cells. For instance, an asymmetric distribution of mitochondria is observed in the production of mouse oocytes, ensuring a high content of mitochondria in the early mammalian development and proper mitochondrial maternal inheritance [45].

In HSCs, an asymmetric distribution of organelles (mainly lysosomes and mitochondria) upon cell division has been associated with distinct cell fate, with the asymmetric distribution of mitochondria correlating with the energetic and metabolic profiles of the progenitor cells [46]. The mitochondrial network dynamic seems to play a major role to ensure this asymmetric distribution in the daughter cells which plays a central role in the stemness maintenance of HSCs. Indeed, upon fission disruption (DRP1 inhibition for instance), the HSCs lose their regenerative capacities and are blocked in a quiescent deregulated state [47]. When inducing the clearance of mitochondria, using the NAD+-boosting agent nicotinamide riboside (NR), researchers have been able to increase HSCs asymmetric divisions and to enhance their stem cell potential in a mouse model [48].

Similarly, the fusion competency of the mitochondrial network is also of utmost importance for the control of mammary stem cell differentiation. Indeed, in a model of epithelial-to-mesenchymal transition in mammary stem cell asymmetric division, the segregation of fused mitochondria close to the cortical membrane ensures the asymmetric distribution of mitochondria in the stem cell progeny. This process ensures the appropriate luminal differentiation of the progeny and the maintenance of the cortical mammary stem cell, while the disruption of the mitochondrial network fused state leads to a symmetric cell division of the progenies undergoing both luminal differentiation [49]. In addition, the proteomic analysis of the basal/cortical and luminal progenitors in mammary epithelial cells reveals a heterogeneous metabolic profile and mitochondria content [50]. Altogether, these studies further support the physiological importance of the mitochondrial asymmetric repartition leading to specific metabolic signature in stem cell progeny, ensuring tissue homeostasis through stemness and differentiation regulation.

Most interestingly, Katajisto and co-workers not only described an asymmetric distribution of mitochondria in human mammary stem cell (hMaSC) progeny but also demonstrated an asymmetry in the age of the distributed mitochondria [17]. For this experiment, the authors used a sequential Snap-tag labeling method. Briefly, the mitochondrial outer membrane protein 25 (Omp25) was first labeled with red fluorescence, and then, after a period of time, the newly produced mitochondria were labeled with green fluorescence, enabling the authors to distinguish “old” mitochondria from “young” mitochondria. Using fluorescence-activated cell sorting (FACS), they were able to sort cells according to their old or young mitochondria content. The authors showed that the cells inheriting a mixture of old and young mitochondria took the path of the differentiation, whereas the cells that retained their stemness properties received almost exclusively young mitochondria [16]. This finding suggests a link between asymmetric distribution of mitochondria based on age and cell fate determination.

Further supporting these results, Adams et al. (2016) reported similar observations in a different cell type, namely T lymphocytes, and in an in vivo setting [39]. In their study, mice infected with *Listeria* exhibited an asymmetric distribution of old and young mitochondria correlated with the cell fate in differentiated versus self-renewing lymphocytes. The differentiated cells showed an enrichment of old mitochondria compared to self-renewing T lymphocytes. The authors demonstrated that this asymmetric mitochondrial distribution modulated cellular metabolism, specifically the balance between catabolism and anabolism, with implications for overall cellular metabolism as explained below. The in vivo part of this study provides evidence that the age-related mitochondrial asymmetric distribution is a physiological phenomenon associated with cell differentiation and not an epiphenomenon linked to in vitro culture [39].

More recently, in 2022, the group of Katajisto pursued the characterization of the so-called “old” and “new” mitochondria and showed that the youngest mitochondria have a typical stem cell metabolism, with few oxidative phosphorylation (OxPhos) activities, low reactive oxygen species (ROS) production, and immature mitochondrial morphology (poorly developed, primitive cristae). Old mitochondria are characterized by a higher level in the electron transport chain (ETC) subunit Rieske iron-sulfur protein (RISP), responsible for the first electron transfer of the complex III. Moreover, a difference in the mitochondrial membrane potential between old and new mitochondria was observed, confirming the difference in OxPhos efficiency [17] (Figure 1). In this comprehensive review, the terms “old” and “new” are consistently employed to describe older, mature, and active mitochondria and younger, less active mitochondria, respectively, in accordance with the existing literature.

This work unveils a potential new layer of regulation for mitochondrial partitioning in the control of stemness. While the metabolic and molecular implications of the asymmetric distribution are discussed in the fifth section, the involvement of mitochondrial dynamics and quality control in the mitochondria asymmetric division are discussed below.

## 4. The Involvement of Mitochondrial Dynamics in Asymmetric Mitochondrial Apportioning

While the mitochondrial segregation reported during lymphocyte differentiation was observed only from the cytokinesis stage onward [39], in the model of mammary stem cells, old and young mitochondria were segregated in the cytoplasm of the mother cell before cell division, with the old ones being perinuclear and the young ones evenly distributed in the cell (Figure 1) [17]. Interestingly, the number of cells inheriting mainly young mitochondria decreases following inhibition of mitochondrial fission by a Drp1 inhibitor [16]. These results suggest that mitochondrial segregation in the mother cell is not only responsible for and leads to asymmetric partitioning but also that this mechanism is dependent on the fission, fusion, and mitophagy machinery (Drp1-dependent).

The impact of mitochondrial dynamics and activity on cell fate has been demonstrated in a recent study [49] focusing on mammary gland human stem cells, both in vitro and in vivo using mice models. The study revealed a molecular mechanism underlying the establishment of mitochondrial segregation in the mother cell during asymmetric division. In this research on a model of epithelial-to-mesenchymal transition in mammary stem cells, an asymmetric division occurs, where the fused mitochondrial network, which is more oxidative, was specifically segregated in the mother cells and subsequently polarized in the differentiated daughter cells. Of note, this segregation of fused mitochondria did not occur during symmetrical division. It is proposed that the fused mitochondria segregation in the mother cell involves a molecular mechanism that includes two key components: mitofusin 1 (MFN1), involved in the mitochondrial fusion process, and the cell polarity complex, consisting of atypical protein kinase C (aPKC), comprising PKCζ and PKCι/λ.

Activation of aPKC by TGFβ1 promotes self-renewal of stem cells and prevents the membrane localization of NUMB, a differentiation marker. Interestingly, the article reports that TGFβ1 treatment leads to membrane relocalization of both MFN1 and fused mitochondria, which is dependent on PKCζ. Indeed, those three actors (MFN1, PKCζ, and NUMB) interact, as demonstrated by co-immunoprecipitation, and the absence of PKCζ results in a shift towards symmetric cell division and a cytoplasmic localization of MFN1-labeled fused mitochondria. The proposed mechanism suggests that TGFβ1 activation leads to the relocalization of fused mitochondria to the cortical membrane, where the PKCζ-MFN1 complex anchors them. Then, the presence of MFN1 close to the membrane and in interaction with the PKCζ would be crucial for PKCζ-mediated phosphorylation of NUMB triggering its dissociation from the cortical membrane, thereby maintaining cells in a stem state [49]. Interestingly, a similar mitochondrial-to-membrane tethering mechanism is present in yeast, involving different molecular actors [44]. This supports a potentially conserved mechanism leading to mitochondrial segregation.

While the mechanisms underlying the asymmetric apportioning of young and old mitochondria in the progeny are still unclear, they seem to be not driven by the mitochondrial membrane potential (ΔΨm). Indeed, the use of a mitochondrial uncoupler did not impact the asymmetric distribution of mitochondria in hMaSC daughter cells but did affect differentiation capacity [16]. The importance of ΔΨm was revealed later in 2022 by the same group, through the increased RISP levels in old mitochondria and decreased RISP abundance in young ones. These results show that ΔΨm, and thus mitochondrial activity, is a driver of differentiation, but is not the source of the skewed distribution [17].

Beside the fusion/fission machinery, mitophagy has emerged as a major contributor to mitochondria asymmetric apportioning in stem cell division. Indeed, in addition to its well-known role in organelle homeostasis and in the response to cellular stress by the PINK1/Parkin pathway (see [51]), mitophagy contributes to determine the type of mitochondria found in a stem cell and a differentiated cell [16,39,52,53]. Both Katajisto and Adams’s studies found a higher mitophagy activity in stem-like cells (SLCs) than in epithelial cells [16] and in T/B cells compared to differentiated resident cells [39], supporting the relevance of mitochondrial clearance for self-renewal capacity. Of note, aged mitochondria colocalized with lysosomes and autophagosomes. Upon treatment with mDivi-1, an inhibitor of DRP1 protein, or with chloroquine, a general inhibitor of macroautophagy, an increase in aged mitochondria in T and B lymphocyte cells is reported, promoting the differentiation of those cells [39]. These results strongly suggest that the phenomenon of asymmetric mitochondrial distribution plays a critical role in directing cell fate, rather than being a mere consequence of the differentiation process. However, as manipulating the actors of mitochondrial dynamics also influences differentiation processes independently of asymmetric mitochondrial apportioning or even of asymmetric division (reviewed in [30]), direct evidence for a driving role of asymmetric mitochondria apportioning in cell fate decision is currently still missing.

Mitochondrial dynamics, which determines whether old mitochondria are degraded or retained, has significant implications for cellular metabolism. Stem cells degrade more their old mitochondria and thus tend to adopt a catabolic metabolism through macro autophagy while differentiating cells that retain their old mitochondria exhibit an anabolic metabolism [39]. The influence of this asymmetric distribution on cellular metabolism, in general, is the focus of investigation in the upcoming section. The objective is to understand the causal relationship between mitochondrial dynamics, including asymmetrical distribution, and cellular metabolic processes, shedding light on the broader impact of mitochondrial dynamics on cellular physiology and function.

## 5. Metabolism of Stem Cells and Progenitor Cells

Preserved by new and healthy mitochondria, stem cell basal metabolism is mainly characterized by the primary use of glycolysis to meet energy requirements rather than OxPhos. On the contrary to the differentiated progeny, stemness is associated with higher lactate production, as well as lower oxygen consumption rates and intracellular ATP content [54]. While OxPhos is more efficient in producing ATP per glucose intake, glycolysis provides ATP more rapidly. The importance of these different metabolic profiles during and prior to the differentiation process has been initially termed the “metabolic state hypothesis” by Prigione et al., 2010 [55].

Until recently, a question that remained persistent was the temporality of this metabolic shift and the causal links that connect it to cell differentiation. Does the metabolic shift drive differentiation, or is it a consequence of it? Much research has since provided concomitant results with the driving model, and even more with the arrival of new studies on iPSCs and the study of the reprogramming phenomenon. In 2006, Takahashi and Yamanaka promoted pluripotency in a differentiated fibroblast cell by the induction of specific factors. These Yamanaka factors, Oct3/4, Sox2, c-Myc, and Klf4 promoted fibroblast dedifferentiation into iPSCs by pushing the reprogramming process [56]. By reversing the chronology of the differentiation, iPSCs provide information about the mechanisms underlying the acquisition and maintenance of stemness from differentiated cells. Studies on iPSCs and data on the mechanism of reprogramming and differentiation are numerous (see the review from [57]). Among these, the acquisition of pluripotency features by somatic cells can be favored through an oxidative-to-glycolytic shift mediated by the induction of glycolic genes by a transcription factor (TF), such as Myc1 [58]. Interestingly, the reprogramming efficiency is enhanced in somatic cells that already predominantly use glycolysis, such as human umbilical vein endothelial cells or keratinocytes [59]. The induction of pluripotent markers in iPSCs occurs after the promotion of glycolytic genes [60], thus supporting the glycolysis driver model. In quiescent HSCs, OxPhos limitation is necessary to preserve stemness and inhibit differentiation [61,62]. These results suggest an essential role of the glycolysis vs. respiration balance in stemness acquisition and maintenance through nuclear reprogramming. While the use of glycolysis is necessary to maintain multipotency in ASC and to induce pluripotency in iPSCs, the use of OxPhos initiates differentiation.

## 6. Benefits and Regulation of Glycolytic Metabolism

The glycolytic metabolism preference of stem cells is closely related to their microenvironment and their activity. Highlighting how glycolysis is sustained and aerobic respiration restricted in stem cells provides insight into the key role of the mitochondria. Previously, these metabolic preferences were mainly explained by two reasons: the glycolysis promotion/OxPhos inhibition under hypoxic conditions, and the provision of metabolic intermediates derived from the glycolysis. Since Katajisto and colleagues’ study, a third reason has emerged: the low level of RISP in young mitochondria. These three reasons and the links between them are discussed below.

First, HSCs, MSCs, NSCs, and even naïve ESCs (prior to implantation) live in a low-oxygen environment [63,64,65]. Under such hypoxic conditions, hypoxia-inducible factor 1 (HIF1), a nuclear heterodimeric transcription factor with two subunits (α & β), preserves stem cells’ pluripotency by promoting anaerobic metabolism. The HIF1 subunits are hydroxylated and degraded under normoxic conditions (2~9% of oxygen) by prolyl hydroxylase and stabilized under hypoxia. HIF1α modulates the metabolism through multiple target genes, including glycolytic genes, and the gene coding for the pyruvate dehydrogenase kinases (PDKs) 2 and 4 [66]. By phosphorylating and consequently deactivating the pyruvate dehydrogenase (PDH) complex, which initiates the tricarboxylic acid cycle (TCA), the PDK prevents mitochondrial respiration as well as mtROS production and promotes glycolysis [67]. In the presence of a PDK inhibitor such as dichloroacetate (DCA), ESCs initiate differentiation and lose their proliferative capacity and pluripotency [68]. By maintaining glycolytic metabolism, the inactivation of PDH through the HIF1 pathway is shown to promote pluripotency and stem cell features. Moreover, the expression of core pluripotency genes such as OCT4, SOX2, and NANOG, involved in self-renewal and stem cell features, is regulated by HIF2α [69]. This is mediated by the HIF2α upregulation of C-terminal binding protein (CTBP) expression, a metabolic sensor acting as transcriptional corepressor or coactivator. Silencing of HIF2α, as well as CTBP in hESCs, results in the loss of pluripotency markers (POU5F1, SOX2, and NANOG), decreased proliferation, along with an upregulation of the early differentiation marker SSEA1 [70].

Not only is glycolysis promoted by HIF1, OxPhos is also inhibited by several pathways in order to not initiate differentiation [71]. Among these, mitochondrial carrier homolog 2 (MTCH2), a downstream actor in the ATM–BID pathway, has been shown to be involved in HSC stemness maintenance [72]. The loss of MTCH2 in HSCs results in enhanced OxPhos and consequently leads to differentiation. However, the MTCH2-mediated OxPhos inhibition mechanism is still not fully understood. Nonetheless, the increase in mitochondrial activity was correlated with an increase in mitochondrial size through hyperfusion. This hyperfusion is related to a restrictive effect of MTCH2 on the translocation of Drp-1 to the mitochondria and thus has a restrictive effect on mitochondrial fission [72]. Thus, the mechanisms of mitochondrial fission and fusion are once again related to metabolism and cell fate and are therefore finely regulated by the stem cell.

Second, stem cells promote glycolysis in order to produce various metabolites further shunted to other pathways. Indeed, glycolysis provides building blocks such as co-factors and substrates of several other biochemical pathways needed for rapid cell proliferation and stem cell activity. During the rapid expansion phase, stem cell proliferation involves the activation of anabolic pathways leading to the synthesis of DNA, lipids, amino acids, etc. [73]. The high glycolysis rate in ESCs and iPSCs allows them to shunt the flow of intermediates such as the glucose 6-phosphate to the pentose phosphate pathway (PPP) used for purine biosynthesis, the acetyl-CoA and dihydroxyacetone phosphate that can be used for lipids synthesis, and the 3-phosphoglycerate for amino acid synthesis through the homocysteine cycle [74]. As expected, higher carbon shunting to the PPP pathway has been reported in cells receiving young mitochondria [17].

Third, the maintenance of glycolysis in stem cells for the reasons mentioned above is also, and importantly, a result mediated by the inheritance of young, immature mitochondria. As previously explained, cells receiving new mitochondria have stem cell properties associated with increased use of glycolysis, whereas cells inheriting old ones initiate the differentiation. New mitochondria have been shown to be depleted in the RISP complex, exhibiting stemness by decreasing their efficiency in ETC and oxidative metabolism [17]. The implication of the loss of RISP on the efficiency of cellular respiration as well as on cell fate had already been demonstrated in previous studies on different cell types. Indeed, antimycin-mediated RISP inhibition results in the expression of pluripotency genes in ESCs such as OCT4, even after treatment to trigger differentiation [75]. Tormos and coworkers had previously found that the RISP knockdown restrains the MSCs’ differentiation into adipocytes. Their results suggested that the RISP-mediated superoxide generation during the Q cycle was essential for the differentiation initiation. When this mtROS production is reduced by the mitochondrial-targeted antioxidants MitoCP, the MSC differentiation is also prohibited [76]. Similarly, Ansó et al. showed in 2017 that the loss of RISP in fetal mouse HSCs by deleting its gene Uqcrfs1 leads to anemia and thus to prenatal death. This pathology was the result of a reduced HSC repopulation capacity through the depletion of myeloid progenitors and erythroid precursors [77]. Overall, these results show that a high RISP level and consequently OxPhos activity is essential for and acts upstream of stem cell differentiation. These findings provide compelling evidence that the maintenance of glycolysis plays a pivotal role in constraining oxidative metabolism while supplying crucial intermediates for various cellular pathways. This dynamic balance between glycolysis and oxidative metabolism ensures a steady supply of energy and essential metabolites to sustain stem cell functions and support their diverse metabolic demands.

## 7. Benefits and Regulation of Oxidative Metabolism

Old mitochondria show a higher level of the RISP subunit in the ETC, associated with OxPhos-enhanced oxidative energy metabolism and an increase in total cellular ROS level [17]. Similar results were obtained in the study of Adams and coworkers, where lymphocyte cells carrying aged mitochondria also showed higher mitochondrial ROS [39]. The production, form, scavenging, and consequences of ROS have been previously reviewed [78,79]. This excessive oxidative stress may lead to a decline in repopulation and exhaustion ability, or even to apoptosis of stem cells [80]. On the contrary, insufficient ROS production in stem cells results in impaired proliferation, as well as reduced stem cell differentiation and self-renewal capacity [81]. The promotion of ROS production through increased mitochondrial fatty acids and carbohydrate metabolism activity induces stem cell differentiation [82]. For instance, a key role of ROS has been revealed in bone marrow injury recovery, in which ROS is needed for the stimulation of HSC proliferation and progenitor cell differentiation into osteoclasts [83]. Therefore, adult stem cells generally require a low basal level of ROS production to preserve quiescence and self-renewal capacity, whereas a moderate increase in ROS production is required before the differentiation [84]. Thus, the delicate equilibrium of the ROS level in the cell implies a fine regulation of the cell redox status through tight regulation of OxPhos activity.

The signaling downstream of the ROS-mediated differentiation is not fully elucidated and varies between stem cell types [85]. Importantly, it should be mentioned that many studies on the implication of ROS on cell fate are based on other ROS sources than those related to mitochondrial metabolisms, such as NOX/ RBOH and external sources [86]. However, some relevant cases can be highlighted, illustrating the tight control of ROS on cell fate decisions. p38-mitogen-activated protein kinase (MAPK) is one of the many pathways that have been shown to be activated by cellular ROS accumulation, leading to a loss of quiescence and exhaustion in various stem cells. ROS-mediated phosphorylation of p38 MAPK in HSCs triggers proliferation and differentiation through the promotion of purine metabolism. Upon intense stress (blood loss, transplantation, etc.) requiring the generation of hematopoietic cells, p38α initiates the proliferation and differentiation of HSCs through the activation of the microphthalmic-associated transcription factor (MiTF). MiTF, once activated, binds to the promoter and activates the transcription of a gene coding for a purine metabolism enzyme, inosine monophosphate dehydrogenase 2 (IMPDH2) [87,88]. Interestingly, although many pathways dedicated to the response to redox imbalance exist (FOXO, P53, etc.), quiescent cells show a higher expression of some of these genes compared to differentiated progeny [89,90]. Concomitant with these results, satellite cells contain less cellular ROS than differentiated cells [90]. This could be explained by the need for stem cells to protect themselves from these ROS, therefore preventing differentiation.

Several other pathways have been identified to regulate stem cells through ROS signaling. For instance, the p53-ROS pathway has been implicated in the regulation of adipogenesis [91], and the ROS-mediated activation of NRF2, a critical transcription factor involved in cellular redox homeostasis, has been shown to regulate neural stem cell (NSC) differentiation [92,93].

While the role of ROS as a second messenger in cell signaling is well known, these results emphasize the prominent role of ROS in the regulation of stem cell differentiation. Thus, it is tempting to hypothesize, following data provided by Katajisto in 2015, Adams in 2016, and Dohla in 2022, that the asymmetric distribution of mitochondria impacts cell fate through a ROS response mechanism (Figure 2). The decrease in the expression of ETC complex III in new mitochondria maintains stem-like qualities in the daughter cells, mediated by the maintenance of the cell’s stable and low redox state. Conversely, increasing the level of RISP expression in the ETCs of old mitochondria drives cell differentiation through the ROS-induced response.

## 8. Epigenetics Changes and Cell Fate

Epigenetics is defined as the reversible modifications of DNA and histones that modify gene expression. By decreasing or increasing the expression of genes involved in stem cell differentiation, the rewriting of the epigenetic pattern within the cell has an essential role in determining cell fate [94,95]. Many studies have shown differences in the epigenetic pattern of stem cells but also between the stem and differentiated state. For example, HSCs exhibit DNA methylation levels that vary with hematopoietic lineage commitment in a very locus-specific manner, favoring or disfavoring genes involved in differentiation into myeloid or lymphoid progenitors [96]. Another relevant and interesting example is that almost one-third of the epigenome of hESCs differs from its counterpart differentiated into primary fibroblasts [97]. Cofactors and substrates involved in these epigenetic processes are derived from metabolites generated in major metabolic pathways, such as the Krebs cycle, the folate cycle, or glycolysis [98]. This tight bidirectional correlation between epigenetics and cellular metabolism has been highly reviewed in the literature [71,99,100] but is still not linked to asymmetric mitochondrial apportioning. Thus, this section aims to highlight how the asymmetric distribution of mitochondria and the related impact on metabolism can influence cell fate through epigenetic modifications.

The increase in glycolysis and the decrease in OxPhos in cells inheriting young mitochondria have an impact on the metabolome of the progenitor cell [16]. Through metabolic changes, it is the metabolome variation that influences the epigenetic mechanisms of the cell (Figure 2). Döhla’s study reports changes in metabolite abundance in the cell population receiving the new mitochondria and maintaining stemness. Thus, related to the decrease in TCA and OxPhos activity, this cell population is characterized by a decrease in the NAD+/NADH ratio (due to decreased electron consumption), as well as a decline in fumarate abundance [17]. This ratio is important for epigenetic processes since NAD+ is required for the activity of class III NAD+-dependent histone deacetylases (sirtuins) and histone deacetylating enzymes [101]. Deactivation of sirtuins due to a low NAD+/NADH ratio increases the expression of genes involved in stem cell pluripotency but also in their glycolytic metabolisms [102,103]. These data are consistent with results obtained in iPSCs. Indeed, while the under-expression and suppression of sirtuin 2 (SIRT2) results in increased expression of genes involved in glycolysis, thus facilitating and increasing the efficiency of cell reprogramming, enhanced expression of SIRT2 results in the differentiation of ESCs [104]. However, due to the reversibility of the epigenetic processes, the effective epigenetic state in the cell is highly dynamic and cannot be restricted to this single aspect. SIRT1, for example, is a sirtuin (also NAD+ dependent) highly expressed in ESCs and contributing to maintain pluripotency. Indeed, the potency of hESCs and HSCs is promoted by SIRT1, which represses by deacetylation the activity of P53 known to induce differentiation [105]. Moreover, when activated, SIRT1 inhibits the differentiation and proliferation of NSCs and MuSCs [98]. In the latter case, differentiation is inhibited by SIRT1-mediated deacetylation of myogenesis-related genes [98]. In addition, loss of pluripotency in ESCs and differentiation of muscle cells is observed upon SIRT1 deficiency [98]. Thus, chromatin acetylation and deacetylation are not strictly related to the promotion of differentiation and self-renewal respectively. These processes are continuous and are complementary to other epigenetic mechanisms, forming a specific pattern of acetylation and methylation adapted to the cell fate (expansion, exhaustion). These results reflect the concept of hyperdynamic chromatin linked to the chromatin remodeling plasticity carried by stem cells [106], unlike other somatic cells whose chromatin is more stable over time.

Another recent study with complementary results is that of Ansó et al., based on RISP gene deletion in HSCs, a situation comparable but not identical to RISP depletion in young mitochondria. In their study, RISP gene deletion also results in a decrease in the NAD+/NADH ratio, while succinate, fumarate, and 2-hydroxyglutarate (2HG) increase in abundance, leading to impairment of the differentiation process [77]. These TCA metabolites are all antagonists of several α-ketoglutarate (α-KG)-dependent dioxygenases, a family of enzymes required for DNA and histone demethylation [107]. Those enzymes, such as histone demethylase (KDMs) and DNA demethylases (TETs), remove methyl groups from the cytosine of DNA and arginine or lysine residues of histones [108]. Accumulation of these metabolites inhibits these enzymes, leading to the maintenance of the DNA and histone methylation pattern and thus modifying gene expression. While Döhla’s study notes only a slight increase in succinate following RISP depletion, it unfortunately does not report on the status of the cellular methylome. Anso’s study, on the other hand, reveals histone and DNA hypermethylation, consistent with the increase in 2HG, succinate, and fumarate, acting as demethylase inhibitors [77]. These two epigenetic processes should be detrimental to stemness preservation in cells inheriting old mitochondria since the NAD+/NADH and succinate/alphaKG ratios are reversed.

Also, the acetyl CoA is involved in the acetylation of histones by donating its acetyl group to the lysine residue, catalyzed by acetyltransferases (HATs) [109]. Thus, the level of histone acetylation follows the concentration of acetyl-CoA. By not impeding the electron flow in OxPhos, old mitochondria produce enough reducing power to not saturate TCA. TCA intermediates should thus be more abundant in cells receiving old mitochondria, which is shown experimentally [17]. As the turnover and utilization of these cofactors and substrates are mediated by the anapleurotic reactions of mitochondrial metabolism, they influence and contribute to histone and DNA epigenetic plasticity and thus to cell fate. DNA hypermethylation, through depletion of the RISP complex characteristic of young mitochondria inheriting cells, could thus repress gene expression involved in progenitor cell differentiation and maintain self-renewal, as it was observed in HSCs (Figure 3) [77].

## 9. Conclusions and Future Research Proposal

The influence of mitochondria on stem cells has already been well illustrated in previous papers, notably with mitochondrial dynamics such as fission/fusion and mitochondrial biogenesis/mitophagy. This work attempts to put emphasis on the importance of the asymmetric inheritance of young and old mitochondria, regulated notably through mitochondrial dynamics, in stem cell division. This review highlights the contrasting characteristics of the two mitochondrial populations, as evidenced in the literature, encompassing their age, maturity, and activity. The emphasis is on the pivotal role played by the asymmetric distribution of these mitochondrial pools in dictating cell fate by influencing both stem cell metabolism and epigenetics (Figure 3). This effect can be achieved by modulating the abundance of key metabolites, such as the production of reactive oxygen species (ROS) and epigenetic modulators, but also by the manipulation of the redox level and cellular energy state. This intricate interplay leads to the activation of diverse pathways that play crucial roles in either maintaining stemness properties or driving cellular differentiation. However, the mechanisms by which young mitochondria maintain stem cell properties while old mitochondria initiate differentiation have yet to be thoroughly investigated, providing exciting avenues for future research.

One important direction for future studies would be to elucidate the molecular, cellular, and tissue-scale mechanics that underlie the segregation of old and young mitochondria. Exploring potential connections between signals from the stem cell niche, which initiate differentiation or ensure self-renewal, and the initiation of asymmetric mitochondrial distribution would be particularly intriguing. A model such as isolated muscle fibers could be used for this purpose, as the satellite cell niche is quite well preserved and stem cell asymmetric division is observable in such a context (for a review, see [110]). Additionally, in line with characterizing the omic differences between old and young mitochondria, it would be valuable to investigate whether RISP is the sole determinant of mitochondrial cell fate variation between young and old mitochondria, impacting its activity. Furthermore, extending the characterization of this phenomenon to other cell types and differentiation models would provide valuable insights.

Continued investigations are necessary to uncover the precise mechanisms underlying asymmetric mitochondrial distribution and its profound impact on tissue homeostasis. Nevertheless, this review highlights potential pathways through which aged/young mitochondria can exert their influence, including the interplay of reactive oxygen species (ROS), metabolism, and epigenetics, which are highly interconnected.

## Figures and Tables

**Figure 1 ijms-24-12181-f001:**
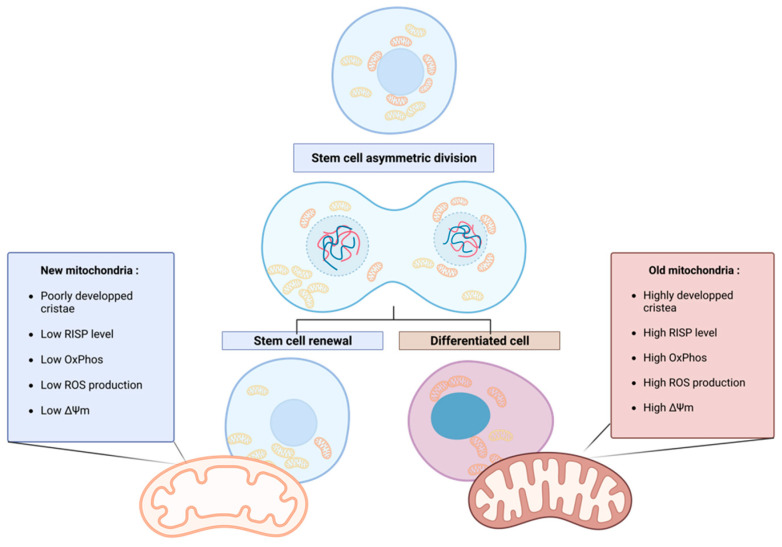
Asymmetric apportioning in human mammary stem-like cells (hMaSCs) and its impact on stem cell fate progeny. Old mitochondria (**right**) are characterized by a higher level of RISP in the ETC than young mitochondria (**left**), resulting in the increased oxidative metabolism of the old mitochondria, reflected in higher membrane potential (ΔΨ) and ROS levels. These characteristics initiate the differentiation of the cell inheriting old mitochondria. On the contrary, in cells receiving young mitochondria, the low oxidative metabolism maintains stem cell properties through enhanced glycolytic metabolism. Figure created with BioRender.com.

**Figure 2 ijms-24-12181-f002:**
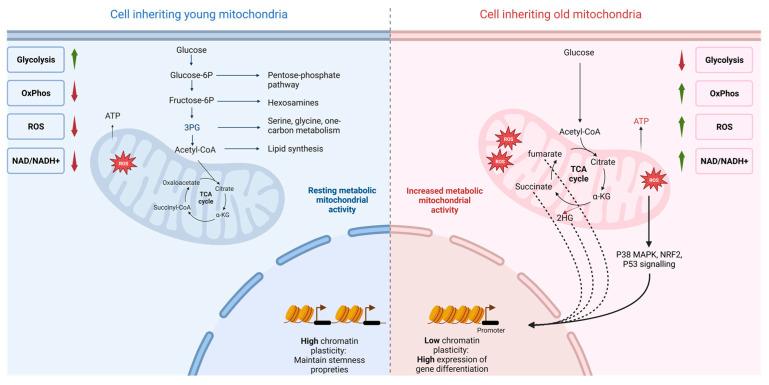
Proposed model for asymmetric mitochondrial apportioning and its impact on stem cell fate progeny metabolism and epigenetic, based on [16,17]. Old mitochondria (**right**) are characterized by a higher level of RISP in the ETC compared to young mitochondria (**left**). This higher oxidative metabolism leads to elevated levels of reactive oxygen species (ROS), a higher ratio of NAD+ to NADH, and increased oxidative phosphorylation (OxPhos) activity. These specific characteristics of old mitochondria play a dual role in initiating cell differentiation. Firstly, through the higher abundance of TCA metabolites, they exert epigenetic effects, potentially modifying gene expression patterns (indicated by the dashed arrow). Secondly, ROS induced by the higher level of OxPhos can activate different signaling pathways that promote cellular differentiation (indicated by the solid arrow) in various stem cell lines. In contrast, cells that inherit young mitochondria experience a reduced oxidative metabolism accompanied by an increase in glycolytic metabolism. This metabolic feature helps maintain stem cell properties by ensuring an abundance of glycolytic intermediates that are utilized in various metabolic pathways crucial for proliferation. Examples of these pathways include the pentose phosphate pathway, one-carbon metabolism, and lipid synthesis. Figure created with BioRender.com.

**Figure 3 ijms-24-12181-f003:**
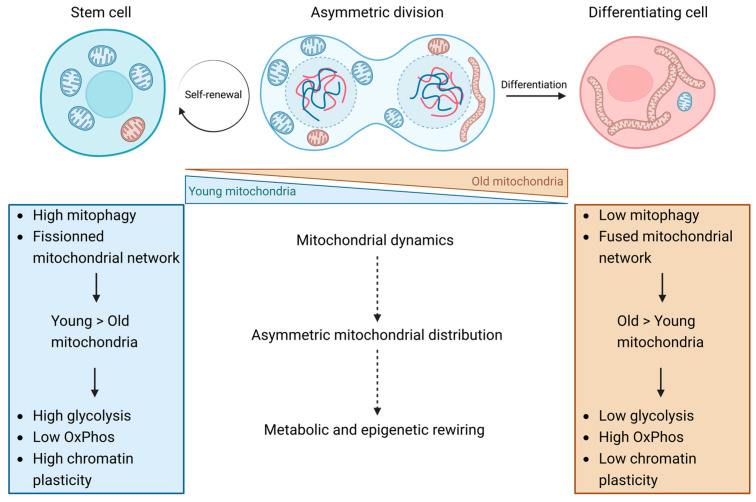
Impact of the mitochondrial dynamics on cell fate through the asymmetric mitochondrial distribution regulation. The asymmetric distribution of mitochondria observed during the asymmetric division of both human mammary stem cells (hMasSCs) and hematopoietic stem cells is of importance in the determination of the progeny cell fate. The daughter cell inheriting a higher ratio of young/old and more fractionated mitochondria are maintained as stem cells, whereas the other progeny, exhibiting a lower ratio of young/old and more fused mitochondria, undergoes differentiation. This asymmetry in mitochondria phenotype and function between the daughter cells defines their metabolic activities and consequently the epigenetic landscapes, thus finally impacting cell commitment to either differentiation or self-renewal. Therefore, the mitochondrial network dynamics, regulated through fusion/fission and mitophagy, is of utmost importance for the regulation of mitochondria asymmetric distribution and drives, through this process, the daughter cell fate. The mitochondrial apportioning is thus a potential direct driver of cell differentiation and contributes to stem cell maintenance. The molecular mechanisms through which the mitochondrial dynamics regulate the asymmetric distribution of mitochondria as well as how this asymmetry impact the metabolism of the daughter cells are only partially unraveled and have only been demonstrated in hMaSCs. Therefore, further studies are required to assess the presence of this process in other differentiation models displaying asymmetric division and to address the underlying molecular mechanisms. Figure created with BioRender.com.

## Data Availability

Not applicable.

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
