# Peer review of "The Key Role of Mitochondria in Somatic Stem Cell Differentiation: From Mitochondrial Asymmetric Apportioning to Cell Fate"

_ijms, 2023, doi:10.3390/ijms241512181_

Round 1

Reviewer 1 Report

1) The authors describe young or old mitochondria, depending on whether it is maintained in the stem cell or in cells that undergo differentiation. When we use old concept, usually means a physiological deterioration, in this case, express a cellular differentiation. Therefore, I propose to use another word. I agree that some of the characteristics of old mitochondria are the mitochondrial degradation described for the aged process, but in this case, it is a characteristic of a cellular change. Can the author propose another word? 

2) In the conclusions section the first paragraph does not describe any real conclusions. Also, the second and third paragraphs provide proposals for future research, therefore, the title of this section should be changed to conclusions and future research proposal.

Author Response

Response to Reviewer 1 Comments

Dear Editors, Dear Reviewer,

It is our pleasure to submit our revised manuscript entitled: “The Key Role of Mitochondria on Somatic Stem Cell Differentiation: From Mitochondrial Asymmetric Apportioning to Cell Fate” (ijms-2509183) for consideration as a publication in International Journal of Molecular Sciences.

We addressed the different comments and issues raised by the Reviewer 1, as detailed below. Thanks to these comments, we believe we managed to clarify some used concepts and to improve our conclusions. We would like to thank the Reviewer 1 for his/her interesting comments.

We thank you very much in advance for considering our revised manuscript for publication.

Sincerely Yours,

  1. Amato, S. Meurant and P. Renard.

Point 1: The authors describe young or old mitochondria, depending on whether it is maintained in the stem cell or in cells that undergo differentiation. When we use old concept, usually means a physiological deterioration, in this case, express a cellular differentiation. Therefore, I propose to use another word. I agree that some of the characteristics of old mitochondria are the mitochondrial degradation described for the aged process, but in this case, it is a characteristic of a cellular change. Can the author propose another word?

Response 1: We totally agree with this comment, the use of “old mitochondria” might indeed be confusing and another word, such as “mature” could be more adapted. Indeed, when speaking of old mitochondria in this review, we in fact mean older mitochondria with more mature phenotype, compared to the younger ones which are more immature, and which are retained in the stem cells. However, in the original paper of Katajisto and colleagues (Katajisto et al, Science 2015), the words “old” and “young” refer to mitochondria retaining two different fluorescent labels administered in a temporally controlled fashion, using either photoactivatable GFP or SNAP-tag approaches. It appeared that the mitochondria retaining the first label administered – and therefore called old mitochondria - were more mature mitochondria transmitted to the daughter cells. For the sake of fidelity, we decided to stick to the word used in the original study but we further emphasized that the “old” concept used here doesn’t correspond to physiological deterioration but to a more mature phenotype of the mitochondria. We thus implemented this precision in the manuscript as such:

Line: 178: “In this comprehensive review, the terms "old" and "new" are consistently employed to describe older mature and more active mitochondria and younger less active mitochondria, respectively, in accordance with existing literature.”

Point 2: In the conclusions section the first paragraph does not describe any real conclusions. Also, the second and third paragraphs provide proposals for future research, therefore, the title of this section should be changed to conclusions and future research proposal.

Response 2: Thank you for this relevant comment. We have improved the conclusion accordingly. The title of the section has been changed to "Conclusion and future research proposal", and the body of the first paragraph has been reorganized and improved to make our point clearer as such:

Line 542: “The influence of mitochondria on stem cells has already been well illustrated in previous papers, notably with mitochondrial dynamics such as fission/fusion and mitochondrial biogenesis/mitophagy. This work attempts to put emphasis on the importance of the asymmetric inheritance of young and old mitochondria, regulated notably through mitochondrial dynamics, in stem cell division. This review highlights the contrasting characteristics of the two mitochondrial population, as evidenced in the literature, encompassing their age, maturity, and activity. The emphasis is on the pivotal role played by the asymmetric distribution of these mitochondrial pools in dictating cell fate, by influencing both stem cell metabolism and epigenetics [Figure 3]. This effect can be achieved by modulating the abundance of key metabolites, such as the production of reactive oxygen species (ROS) and epigenetic modulators, but also by the manipulation of the redox level and cellular energy state. This intricate interplay leads to the activation of diverse pathways that play crucial roles in either maintaining stemness properties or driving cellular differentiation. However, the mechanisms by which young mitochondria maintain stem cell properties while old mitochondria initiate differentiation have yet to be thoroughly investigated, providing exciting avenues for future research.”

We hope and believe that this paragraph is now more conclusive and allows the reader to better appreciate this work.

Reviewer 2 Report

With a lot of interest, I carefully read the review entitled “The key role of mitochondria on somatic stem cell differentiation: from mitochondrial asymmetric apportioning to cell fate”. This review summarises the current knowledge regarding the asymmetric distribution of mitochondria, the fusion, fission and mitophagy, the involvement of mitochondrial age and dynamics and quality control and cell fate. 

The article is very well written and the supporting figures are very helpful. This reviewer would like to congratulate the authors for writing such a nice and interesting review.

Just one typo:  Line 31 opening bracket is missing

Author Response

Response to Reviewer 2 Comments

Dear Editors, Dear Reviewer,

It is our pleasure to submit our revised manuscript entitled: “The Key Role of Mitochondria on Somatic Stem Cell Differentiation: From Mitochondrial Asymmetric Apportioning to Cell Fate” (ijms-2509183) for consideration as a publication in International Journal of Molecular Sciences. We are really glad to read the positive comments on our submitted review manuscript and we thank the Reviewer 2 for those.
Additionally, we addressed the remaining minor typo.

We thank you very much in advance for considering the revised version of our manuscript for publication.

Sincerely Yours,

I. Amato, S. Meurant and P. Renard.

Reviewer 3 Report

Authors present a well-written review article about mitochondrial asymmetric apportioning between stem cells and differentiated cells.However, some of the information presented is still pretty speculative, so I would ask to express some of the statements more cautiously. For instance, it is not really clear whether asymmetric apportioning is the reason or the consequence of cell differentiation. Please discuss this question more extensively.

Figure 2: information presented in the lower part of the figure- “open chromatin maintain stemness properties” and the other way round. This is very general, and nowhere discussed in the text. So please discuss in the text or leave away

 Figure 3: same comments as above- what is the reason, what is the consequence?

 Line 114-115: this is no sentence, is it?

 Line 261: please explain abbreviation OxPhos, when using for first time

Author Response

Response to Reviewer 3 Comments

Dear Editors, Dear Reviewer,

It is our pleasure to submit our revised manuscript entitled: “The Key Role of Mitochondria on Somatic Stem Cell Differentiation: From Mitochondrial Asymmetric Apportioning to Cell Fate” (ijms-2509183) for consideration as a publication in International Journal of Molecular Sciences.

We addressed the different comments and issues raised by the Reviewer 3, as detailed below. Thanks to these comments, we believe we managed to clarify the message of the review and to improve our conclusions. We would like to thank the Reviewer 3 for his/her relevant comments.

We thank you very much in advance for considering our revised manuscript for publication.

Sincerely Yours,

I. Amato, S. Meurant and P. Renard.

Point 1: Authors present a well-written review article about mitochondrial asymmetric apportioning between stem cells and differentiated cells. However, some of the information presented is still pretty speculative, so I would ask to express some of the statements more cautiously. For instance, it is not really clear whether asymmetric apportioning is the reason or the consequence of cell differentiation. Please discuss this question more extensively.

Response 1: We agree with this relevant comment of reviewer 3, which addresses a pivotal concern in the field of stem cell research: the necessity to discern between the driving factors and the resulting consequences of differentiation. What we propose in this review is still speculative notably due to the relatively recent feature of this research area that is currently under investigation and where extensive work remains to be done.  

In a more general way, the role of mitochondria in cell differentiation has been long debated to determine if the oxidative shift and maturation of mitochondria as well as the regulation of mitochondrial network drives or is a consequence of cell differentiation. Nowadays, it is commonly accepted that mitochondrial would more drive the differentiation. Similarly, we propose that the recently described asymmetric apportioning of mitochondria could be an upstream regulator of cell fate determination, based on observations described in the literature. First, the fact that this segregation of young and aged mitochondrial pools occurs in the mother stem cell, prior to differentiation, implies that the asymmetric distribution is not solely dependent on the differentiation process itself. Second, studies from other researchers have demonstrated that interfering with this distribution, particularly by manipulating mitochondrial dynamics and the cleavage of aged mitochondria, leads to significant alterations in the differentiated phenotype of the cell. However, it is true that manipulating the actors of the mitochondrial dynamics affect cell differentiation in a broader context than asymmetric cell division. To further support a decisive role of asymmetric mitochondrial distribution in directing cell fate, a more complete picture of this phenomenon is required to identify specific molecular target in order to examine the effect of their knock down on cell fate.

In order to highlight this point, we have both nuanced several sentences (lines 58, 125, 209, the nuanced term is underlined) and added a short paragraph (line 258). We hope that this clarification strengthens the coherence of our review and provides a comprehensive perspective on the intricate interplay between mitochondrial dynamics and stem cell fate determination.

Line 58: “Two recent studies from the group of Katajisto in 2015 and 2022 suggest that the answer to this cell fate decision could reside in the metabolic impact of the mitochondrial dynamics and their asymmetric apportioning in stem cell [16] [17].”

Line 125: “In HSC, an asymmetric distribution of organelles (mainly lysosomes and mitochondria) upon cell division has been associated with distinct cell fate, with the asymmetric distribution of mitochondria correlating with the energetic and metabolic profiles of the progenitor cells [46].”

Line 209: “This work unveils a potential new layer of regulation for mitochondrial partitioning in the control of stemness.”

Line 258: “These results strongly suggest that the phenomenon of asymmetric mitochondrial distribution plays a critical role in directing cell fate, rather than being a mere consequence of the differentiation process. However, as manipulating the actors of mitochondrial dynamics also influences differentiation processes independently of asymmetric mitochondrial apportioning or even of asymmetric division (reviewed in [30]), direct evidence for a driving role of asymmetric mitochondria apportioning in cell fate decision is currently still missing.”

Point 2: Figure 2: information presented in the lower part of the figure- “open chromatin maintain stemness properties” and the other way round. This is very general, and nowhere discussed in the text. So please discuss in the text or leave away

Response 2: We would like to express our gratitude to reviewer 3 for this insightful comment. In our review, we propose the hypothesis that the asymmetric distribution of mitochondria, acting as a driver of differentiation and self-renewal, differentially impacts the metabolism of cells inheriting these mitochondrial pools. We suggest that this metabolic shift could, in turn, influence the epigenetics of cells receiving aged mitochondria, particularly by modulating the chromatin state of the cell. But, indeed, the use of “open” or “close” chromatin is not appropriate in this case and we should rather consider the plasticity of chromatin remodeling carried out by stem cells, which differs from other somatic cells with more stable chromatin over time. Consequently, we have revised the figures and replaced the terms "more open chromatin" and "more closed chromatin" with "higher chromatin plasticity" and "lower chromatin plasticity" respectively.

This refinement in terminology better reflects the dynamic nature of chromatin remodeling in stem cells and enhances the clarity of our review's key concepts.

Point 3: Figure 3: same comments as above- what is the reason, what is the consequence?

Response 3: As replied above, we totally agree that this precision was not brought forth and we thus nuanced in the text as mentioned hereabove and we also modified the legend of the figure 3 accordingly to make the message of the figure clearer but still nuanced:

Line 532: “The mitochondrial apportioning is thus a potential direct driver of cell differentiation and contribute as well to stem cell maintenance.”

Point 4: Line 114-115: this is no sentence, is it?

Response 4: Indeed, the sentence should be improved and clarified. We thus modified the sentence accordingly:

Line 114-115: “While it is largely accepted that mitochondria, and more particularly mitochondrial dynamics, plays a prominent role in stem cell behavior, it is only recently that a potential contribution of asymmetric apportioning of mitochondria emerged as a new player in stem cell fate determination.”